# Liver Transplantation for Unresectable Colorectal Liver Metastasis: Perspective and Review of Current Literature

**Todd J. Robinson, Kaelyn Cummins and Allan Tsung \***

Department of Surgery, University of Virginia, 1215 Lee Street, Charlottesville, VA 22908, USA; ekx6sj@virginia.edu (T.J.R.)
\* Correspondence: allantsung@virginia.edu; Tel.: +1-434-924-2158

**Abstract:** The treatment of unresectable colorectal liver metastasis (CRLM) has previously been limited to palliative chemotherapy. Traditionally, the role of liver transplant has not been associated with sufficient survival to justify a patient undergoing a major operation with the associated requirement for postoperative immunosuppression. With improvements in chemotherapy options, a certain subset of patients can experience stable disease for years, which has prompted investigation into the role of liver transplant in these patients. Several recent studies have shown promising results in well-selected patients, with posttransplant survival approaching that of liver transplant recipients for other diseases. Here, we present a review of the data and current protocols for liver transplant for unresectable CRLM.

**Keywords:** liver transplant; liver tumor; oncology

## 1. Introduction

Colorectal cancer (CRC) is the third most commonly diagnosed cancer and the second most common cause of cancer mortality worldwide [1]. Because of current screening guidelines, it is frequently diagnosed at early stages, but approximately 20% of patients have metastatic disease upon diagnosis [2], with the liver being the most frequent site of distant CRC metastases. However, only 20% of patients with CRLM are candidates for resection upon diagnosis [3,4], although this rate may be improved with preoperative systemic or local therapy [5–7].

As such, the current standard of care for unresectable CRLM includes systemic chemotherapy, targeted therapy, and immunotherapy based on tumor characteristics (MMR/MSI status, KRAS/NRAS/BRAF, and HER2 mutation), as well as locoregional therapies for liver metastases (radioembolization, hepatic artery infusion pump) [8,9]. For patients with isolated unresectable liver metastases, conversion to resectability may be a reasonable goal [9]. Recommendations put forth by the European Society for Medical Oncology suggest that the patient's response to neoadjuvant therapy should be evaluated early after the initiation of therapy to determine resectability, although some patients with technically resectable disease may not require neoadjuvant systemic treatment [10]. Patients who present with unresectable liver metastases and who successfully undergo conversion therapy followed by resection have worse outcomes than do patients who present with initially resectable disease but have better outcomes than do matched patients who do not undergo resection [10,11].

Surgical resection is based upon the principles of maintaining an adequate liver remnant, arterial and portal venous supply, hepatic venous drainage, and biliary drainage. Surgical resection is the preferred method to treat metastatic disease, but a combination of resection and local ablative therapies may be combined. Hepatic artery infusion pump has also been demonstrated to be an effective therapy, increasing both overall survival (OS) and disease-free survival (DFS) after hepatic metastasis resection, as well as converting

unresectable to resectable disease [12]. Unfortunately, a significant fraction of patients who undergo resection and other locoregional therapies ultimately have disease recurrence [13].

Given the low number of patients who qualify for surgical resection of liver metastases and the high rate of cancer recurrence after resection, alternative strategies are needed to optimize outcomes in these patients. Liver transplant (LT) is one possible option that is garnering more attention [14–19] in treating unresectable CRLM.

## 2. Historical Perspective

The first described liver transplants for CRLM occurred in the early 1960s and 1970s [20,21]. Outcomes of LT for unresectable CRLM from this time to the mid-1990s were poor, with the 1- and 5- year survival being 62% and 18%, respectively [22]. The practice was essentially abandoned, as LT was difficult to justify for unresectable CRLM given the high rate of tumor recurrence, low patient survival rates, shortage of donor allografts, and the requirement for immunosuppression in patients with malignancy [23].

Over time, however, outcomes improved in patients transplanted for other indications. Improved technical expertise of transplant surgeons led to the decline in technical complications of LT, and improvements in immunosuppression protocols following transplant also contributed to overall survival. As a result, survival after LT improved by 20–30% between the 1990s and 2010s [24]. Additionally, techniques to improve patient selection were advanced, including imaging technology to identify extrahepatic disease and genetic testing to detect lymph node micrometastases [25]. Interest in the possibility of applying LT for unresectable CRLM was accordingly renewed.

## 3. The Secondary Cancer (SECA) Trials

Norway in the late 2000s was uniquely situated in the LT world, with a relative abundance and even surplus of deceased donor livers. The average wait time for LT was less than a month, and there was a system in place to share donor livers with other Scandinavian countries [24]. Despite the increasing number of LTs performed in Norway and Scandinavia, there remained an adequate amount of donor livers to expand the indications of LT to the treatment of secondary malignancy of the liver: in this case, unresectable CRLM. In this setting, the group at University of Oslo devised a protocol for transplant in these patients.

The first SECA trial was published in 2013 [24]. This was a prospective trial aimed at investigating overall survival after LT for unresectable CRLM. Twenty-one patients were included in the final trial analysis. Patients were selected on the basis of having liver-only disease, an Eastern Cooperative Oncology Group (ECOG) performance status of 0 or 1, undergone excision of the primary tumor, and received at least 6 weeks of chemotherapy. Patients were excluded from the trial if they had experienced weight loss of greater than 10%, had other malignancies, or had standard contraindications for LT. Each patient underwent a staging computed tomography (CT) scan at the time of LT. If this was negative for extrahepatic malignancy, these patients underwent a staging laparotomy where hilar lymph nodes and adjacent tissue were sent for frozen section analysis for extrahepatic malignancy. If this was negative for malignancy, the patient then went on to undergo LT. Patients underwent immunosuppression induction with basiliximab and methylprednisolone, followed by maintenance immunosuppression with sirolimus, mycophenolate, and prednisone. No adjuvant chemotherapy was administered, and surveillance imaging was performed every 3 months for the first year and then every 6 months thereafter.

Of the original twenty-five patients enrolled, one patient developed lung metastasis in the preoperative period, and an additional three patients had lymph node metastases on staging laparotomy. As such, 21 patients underwent LT. Explant pathology indicated that 16 of the 21 patients had progressed on first- or second-line chemotherapy. The overall survival at 1, 3, and 5 years, respectively, was 95%, 68%, and 60%. The group noted that 19 of 21 patients had recurrent disease during the study period, with a median time to

recurrence of 6 months. The most common locations of recurrence were in the lung (*n* = 7) and liver allograft (*n* = 7).

Tumor characteristics associated with worse survival included a largest tumor diameter of greater than 5.5 cm, a carcinoembryonic antigen (CEA) level greater than 80 μg/L, and progressive disease at the time of LT, with a combination of these factors predicting even worse survival. Although the recurrence rate in the study cohort was high, the recurrence pattern was such that most patients had disease amenable to resection or ablation therapies and that the 5-year OS exceeded that of most redo LT recipients at the time. The authors concluded that while LT for unresectable CRLM was not ready for widespread adoption, these outcomes warranted development of robust selection criteria and further investigation.

The second SECA trial was published in 2020 and was focused on the development of these selection criteria for patients with unresectable CRLM who would benefit from LT [26]. Selection criteria were further narrowed to include those who had no tumor greater than 10 cm and for those with multiple tumors, (e.g., more than 30), each being less than 5 cm. Additionally, they required a 10% or greater response to systemic chemotherapy and that LT be deferred until the patient was at least 1 year out from the initial diagnosis. If there was less than 10% response to chemotherapy, then inclusion was allowed for patients who had at least a 20% response to transarterial chemotherapy embolization or transarterial yttrium-90 administration. In addition to the exclusion criteria for SECA I, they also excluded patients with BMI > 30. This cohort of fifteen patients included up to T4 tumors (*n* = 1) and one patient with KRAS mutation. The OS at a median of 36 months at 1, 3, and 5 years was 100%, 83%, and 83%, respectively. The disease-free survival at 1, 2, and 3 years was 51%, 44%, and 35%, respectively. The survival from time to relapse at 1, 2, and 4 years was 100%, 73%, and 73%, respectively. Most of the recurrences occurred in the lung (five patients), were small, and were amenable to surgical resection. Factors associated with worse survival were consistent with the results from the SECA I trial and included CEA >80 μg/L, size of largest lesion > 5.5 cm, progressive disease on chemotherapy, and less than 2 years from resection of primary tumor to LT (Table 1). They noted a marked decrease in OS and DFS for patients with a Fong clinical risk score (FCRS) of >2 [27]. They also noted worse overall survival for patients with higher levels of PET-FDG uptake.

**Table 1.** The Oslo score. For each criterion met by the patient pretransplant, 1 point is assigned. Scores range from 0 to 4 points [26].

| |
|---|
| Largest hepatic tumor > 5.5 cm in diameter |
| CEA level > 80 μg/L |
| Time from resection of primary tumor to LT < 2 years |
| Disease progression on chemotherapy |

The authors of the SECA trials also studied LT for unresectable CRLM in a group of patients (labeled "arm D") who did not meet the strict inclusion criteria of SECA II [28]. Ten total patients were included in this cohort. These patients had a history of previous extrahepatic metastasis (pulmonary metastasis: one resected and one possible; prior resected ovarian metastasis), one patient had papillary thyroid carcinoma, and five patients had <10% response to chemotherapy and progressive disease on third-line chemotherapy. Seven patients had moderately or poorly differentiated histology, and one patient had signet ring histology. Three patients with KRAS mutation were included in the cohort. All patients had heavy tumor burden in the liver, with a median of nine lesions and a 60 mm maximum lesion diameter on explant pathology, with preoperative imaging indicative of the same. All patients underwent LT from an extended criteria donor (donors with brain tumor, advanced age, 80% steatosis, bladder cancer, hepatitis B surface antigen positive, significant transaminitis). The median FCRS was three, and the Oslo score was 1. The median DFS and OS were 4 and 18 months, respectively, which were much worse than those reported in the SECA II trial. As seen in the SECA I/II cohorts, the majority of

recurrence occurred in the lung. The authors concluded the results were poor after LT in this "expanded" criteria subset of recipient and donor pairs.

The Oslo group also published 10-year follow-up data on the original cohort of patients from the SECA I trial [29]. In this paper, all patients had recurrence of disease. The median DFS was 10 months, with the lung being the most common site of recurrence. These tumors in general were slow-growing and in most cases resectable. They reported a 5- and 10-year OS of 43.5% and 26.1%, respectively. Their results validated the Oslo score, noting that for patients with a score of either 0 or 1 the 5- and 10-year OS was 75% and 50%, respectively. The authors estimated a 1–3% increase in the number of waitlisted patients should unresectable CRLM be added to the deceased donor waitlist. In additional 15-year follow-up data of the SECA I and II cohorts, they described the impact of metabolic tumor volume (MTV) from PET-CT on survival. They compared low MTV (<70 cm$^3$) vs. high MTV (>70 cm$^3$) and noted improved OS, DFS, and postrecurrence survival (PRS) in the low-MTV group [30].

The SECA patients who underwent LT were also compared to a cohort of 184 patients who had undergone liver resection for CLRM in Padua, Italy [31]. They noted a 5-year OS of 69% in the LT group compared to 14.6% in the resection group for patients with a low Oslo score (<3) and a high tumor burden score (>8). These data are somewhat difficult to interpret given that the LT patients would have automatically been excluded from any study on the efficacy of resection, as the entire point of LT in the setting of CRLM is targeting patients who are deemed unresectable. Furthermore, given the requirement for a year of stable disease prior to LT, the patients in the LT group inherently had a tumor biology that was more stable, which would be in contrast to the spectrum of disease aggression if an all-comers cohort of patients undergoing resection in this setting were considered. Still, the potential survival benefit offered by LT in the setting of CRLM is remarkable, especially when compared to the 30–58% overall survival in resectable CRLM [32,33], which is even lower for all-comers with stage 4 disease [34].

In 2023, the Oslo group reported the 15-year follow up on a pooled cohort of patients from the SECA-1, SECA-2, and RAPID trials [35]. They analyzed the OS and RFS in a total of 61 patients who underwent LT for unresectable CRLM during the study period. Again, they confirmed the validity of the Oslo score as a prognostic indicator, with an OS at 5 and 10 years of 88.9% for both in patients with an Oslo score of 0. The 5- and 10-year survival for patients with an Oslo score of 3 or 4 was 8.3% and 0%, respectively. They also noted that patients with primary tumors in the ascending colon, those with a tumor burden score of 9 or higher [36], and those with nine or more liver lesions were associated with a lower OS.

## 4. Consensus Guideline Development

In response to these advancements, the International Hepato-Pancreato-Biliary Association (IHPBA) published consensus guidelines to address the relevant facets of LT for unresectable CRLM, including patient selection, protocols and screening in anticipation of LT, and graft selection and allocation [37] (see Figure 1 for the treatment algorithm) [37].

The consensus guidelines begin with patient selection. The primary tumor must be resected with clear margins. The tumor must not be undifferentiated or have signet ring histology. N2 nodal disease was considered a relative contraindication, although in the setting of patients with late (>1 yr) metachronous liver metastasis without local nodal recurrence, this was not thought to be as significant. High-quality staging imaging is necessary, including PET-CT with measurement of metabolic tumor volume and total lesion glycolysis. Patients with extrahepatic disease should not undergo LT, and systematic porta hepatis nodal sampling prior to LT should be undertaken. Molecular testing should be done, and any patient with BRAFV600E mutation should not undergo LT. Those with RAS mutations should not be excluded on this basis alone if favorable biological factors are present. Those with microsatellite instability high or deficient mismatch repair should receive immunotherapy instead of LT.

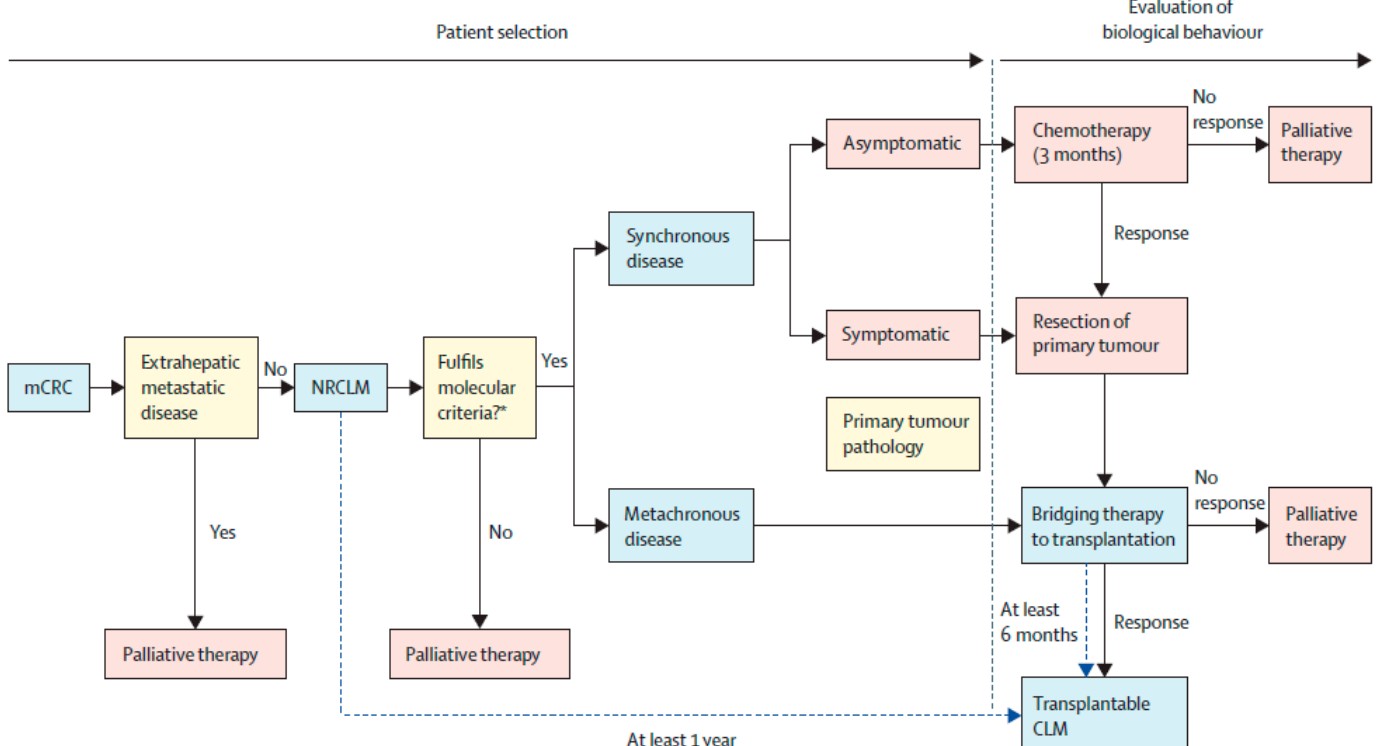

**Figure 1.** A proposed algorithm for the treatment of unresectable colorectal liver metastasis. Used with permission from Bonney et al. [37]. CLM—colorectal liver metastasis; mCRC—metastatic colorectal cancer; NCRLM—nonresectable colorectal liver metastasis.

Patients should receive at least 6 months of fluorouracil-, oxaliplatin-, or irinotecan-based chemotherapy and demonstrate a response of 30% or greater based on the RECIST criteria or have stable disease with response according to the Chun criteria (attenuation, tumor–liver interface, peripheral rim of enhancement) [38]. Those with CEA > 80 μg/L and rising should be excluded from LT, while a level >80 with a downtrend can be considered a relative contraindication. The response to chemotherapy should be observed for at least 6 months, with an interval from diagnosis of unresectable CRLM to LT of >1 year. The order of therapy for asymptomatic disease should allow for neoadjuvant chemotherapy, resection of the primary tumor, and then bridging therapy for LT. In symptomatic disease, the primary tumor should be resected first followed by adjuvant/bridging chemotherapy. CRLM should always be assessed for resectability. If deemed resectable according to traditional principles of liver surgery, there is no indication for LT. For unresectable disease, metachronous disease, or recurrent disease after initial resection that is deemed unresectable, patients should undergo bridging chemotherapy with observation periods as described above.

Graft selection should be at the transplanting center's discretion, but due to waitlist constraints and graft access, it may be necessary to utilize a marginal or extended criteria donor liver with or without novel graft-preservation techniques. Novel procedures such as the RAPID technique (resection and partial liver segment transplant with delayed total hepatectomy) can be employed at centers with sufficient surgical expertise [39]. Living donation for liver transplant (LDLT) should also be considered at centers with well-developed programs. After transplantation, recommended immunosuppression regimens include steroid induction with an interleukin 2 receptor antagonist (e.g., basiliximab), maintenance on a calcineurin inhibitor (tacrolimus) and an antiproliferative (mycophenolate mofetil) followed by replacement of the calcineurin inhibitor entirely with an mTOR inhibitor (everolimus or sirolimus), or a decreased dose of tacrolimus in combination with mTOR inhibitor.

Adjuvant chemotherapy is not recommended at the present time. Recurrences should be managed with surgical resection when able. If systemic therapy for unresectable disease is needed, immunosuppression should be adjusted accordingly. Overall 5-year survival should target a goal of 50% or greater to balance the risk of transplant, risk to the potential living donor, and resource utilization (including deceased donor allograft utilization in the setting of a limited supply). Outcomes should be tracked in clinical trials, an international database, or both.

## 5. Living Donor Liver Transplantation (LDLT) for Unresectable CRLM

LDLT affords the opportunity for transplant in the setting of unresectable CRLM in areas of the world where there are not a sufficient number of deceased donor allografts to meet the demand for transplant. This was described recently in two separate studies. At three centers in North America adhering to the IHPBA guidelines described above, a pooled cohort of donors and recipients was retrospectively analyzed [40]. All potential recipients underwent preoperative PET/CT to assess for tumor progression, and patients with progression were excluded. The living donors consented appropriately, and the focus was on the high likelihood of extrahepatic recurrence of disease based on the SECA trial results. All recipients underwent staged exploration for extrahepatic disease prior to beginning the living donor portion of the operation. Of the 91 potential recipients evaluated, only 12 (13%) were candidates for LT. Two of these patients had an elevated MELD score (chemotherapy-induced liver injury/cirrhosis) and as such, were able to undergo deceased donor liver transplantation (DDLT). The remaining 10 patients underwent LDLT. Of the cohort, nine patients had synchronous disease, and one patient developed metachronous disease 16 months after diagnosis of the primary tumor. Eight of the patients had left-sided or rectal cancer. Six patients had primary tumors with T3 disease, and four had T4a disease. Three patients had poorly differentiated disease on pathology. The median time from diagnosis to LT was 1.7 years. Prior treatments included liver resection ($n = 4$), hepatic artery infusion pump ($n = 3$), and tumor ablation ($n = 3$), and all patients were treated with chemotherapy (median of 22.5 cycles). As stated above, all patients had sustained radiographic or biochemical (measuring CEA levels) response. Eight patients had bilobar disease, and two had a recurrence after prior resection. The median CEA level at the time of transplant was 7.7. The median Fong clinical risk score was 2.5, and the median Oslo score was 1.5. Patients with BRAF V600E mutation were excluded; however, there were notably patients with KRAS, TP53, SMAD4, and BRAF d549G mutations.

Nine patients had viable tumor on explant, including one patient with a positive hilar node that was not noted on initial exploration. Induction immunotherapy included steroid/basiliximab followed by tacrolimus, mycophenolate mofetil, and prednisone. Patients were transitioned to an mTOR inhibitor at approximately 6 months after LT. One of the donors had a subcutaneous hematoma; otherwise, donor complications were mild, and the length of stay was 6 days. Complications among the recipients included one bile leak, one case of rejection, one patient with postoperative ileus, one perihepatic infection, and one case of hepatic arterial thrombosis which was successfully salvaged with return to the OR for thrombectomy. At a median of 1.5 years of follow-up, three patients had recurrence (including peritoneal metastasis in the patient with positive hilar node). These patients were treated with systemic chemotherapy, and one patient had died.

In a review of 14 patients who underwent LT for CRLM in Rochester and Cleveland Clinic, explant liver specimens were examined for residual tumor [41]. Eight patients (57.1%) had undergone LT for unresectable disease, while the remaining six (42.9%) had undergone LT in the setting of liver failure precipitated by chemotherapy and other locoregional therapies. Prior to LT, seven patients (50%) had complete radiographic response, yet eleven patients (78.6%) had residual viable tumor on final pathology despite the radiographic response. Furthermore, nine patients (64.2%) had undiagnosed metastasis not detected on preoperative imaging in the explant. This study highlights the high prevalence

of occult or disappearing metastasis in the liver and lends credence to the idea of total hepatectomy and LT as a treatment option in this setting.

Rajendran et al. recently published an early experience with LDLT for unresectable CRLM [42]. They compared three groups: those who underwent LT, those who converted to resectable disease and subsequently underwent hepatic resection, and those who were treated with palliative chemotherapy. In total, seven of the eighty-one patients referred underwent LT, twenty-two patients underwent resection, and forty-eight received palliative chemotherapy. The inclusion criteria were similar to those described prior, although the authors specified excluding patients with macrovascular invasion and coexisting HIV/HVB/HCV infection. The group also performed staging exploratory laparotomy one week prior to the planned LDLT. Three patients in the LT cohort had previous placement of a hepatic artery infusion pump (the time between insertion and LT was 14.6 to 25 mo). Of the LT group, six patients were on first-line chemotherapy and had completed a median of 20–60 cycles at the time of transplant, and two had previously undergone resection. The median time from assessment to LT was 15.4 mo. The Oslo score was 0–2 for LT patients. Unfortunately, two patients developed recurrence 3.3 mo after LDLT. The median duration of assessment to follow-up was 29.7 months. One patient who underwent LT died at 39.4 months of recurrent disease; eight (36%) of the resected patients died, all secondary to disease progression; and twenty-eight (58%) of the control patients died due to disease progression. OS at 1 and 3 years from time of initial assessment was 81.1 and 15.9% for the palliative chemotherapy population, 100 and 58.8% for the resected population, and 100 and 100% for the transplanted population, respectively. The recurrence-free survival (RFS) in the LT group at 1 and 3 years was 85.7% and 68.6%, respectively. Of the two patients in the LT group who had recurrence, one had lung metastasis and the other had intra-abdominal lymph node metastasis.

## 6. Subsequent Work

Several centers in the US now offer LT for unresectable CRLM, as summarized by Sasaki et al. [43]. Based on the UNOS database from December 2017 to March 2022, 46 patients underwent LT for unresectable CRLM. This work was mainly done at five centers. Twenty patients were able to undergo DDLT (albeit with marginal allografts) due to preoperative liver failure and with a high enough MELD to attract a reasonable quality organ from the waitlist, while 26 patients underwent LDLT. The 1-, 2-, and 3- year OS after LT was 89, 60.4, and 60.4%, respectively. The 1-, 2-, and 3-year DFS after LT was 75.1, 53.7, and 53.7%, respectively. Interestingly, while the MELD score at the time of transplant was not significantly different between the LDLT and DDLT groups, those in the DDLT group had a median total bilirubin level of 2.7, which was significantly worse than that of patients in the LDLT group. This finding would have excluded many of these patients from the SECA trials, and at many centers, such significant liver dysfunction would exclude them from further chemotherapy as well. The survival analysis in this study is confounded by the presence of two distinct phenotypes of LT patients in the setting of CRLM: those who have unresectable disease, for which there are increasingly clear selection criteria with a growing body of evidence of the survival benefit in this setting, and those who undergo LT for liver failure in the setting of CRLM but who may or may not meet the definition of unresectable. The latter group is neither well-defined nor well studied. The study was limited by the retrospective nature, and being from a national database, center-specific protocols, inclusion criteria, and disease pattern analysis could not be performed. Nonetheless, it highlighted a growing activity in the US and overall encouraging outcomes. The authors emphasized the need for a CRLM registry and unified protocols to treat patients with unresectable disease, as well as a need to address issues related to liver allograft allocation to allow the appropriately selected patients access to suitable organs.

## 7. Future Directions

The Oslo group are continuing to study this patient population, and the SECA-III trial (NCT03494946) is forthcoming. There are other randomized trials in process, including TRANSMET (NCT02597348), SOULMATE (NCT04161092) [44], and EXCALIBUR (NCT04898504, NCT04840186) (Table 2). There are also several ongoing prospective trials [19] which will further define selection criteria. These trials should also provide the evidence needed to advocate for change in liver allocation policy.

**Table 2.** Ongoing clinical trials evaluating liver transplantation outcomes.

| Trial | Type | Design | Location | Primary Outcomes | Secondary Outcomes |
|---|---|---|---|---|---|
| TRANSMET | Randomized, controlled | Chemotherapy followed by LT vs. chemotherapy alone | Multicenter; France | OS at 5 years | OS at 3 years, DFS, PFS, recurrence, QOL |
| SECA-III | Randomized, controlled | LT vs. other treatment options including chemotherapy, TACE, or SIRT | Single center; Norway | OS at 2 years | None listed |
| SOULMATE | Randomized, controlled | LT from extended criteria donors + best-established treatment (BET) vs. BET alone | Multicenter; Sweden | OS at 5 years | OS at 2 years, median OS, PFS, recurrence-free survival, QOL, QALY |
| Excalibur 1 | Randomized, controlled | LT + chemotherapy vs. HAI/FUDR + chemotherapy vs. chemotherapy alone | Single center; Norway | OS at 2 years | QOL, 30-/90-day morbidity/mortality |
| SECA-II, Arm D | Nonrandomized, controlled | LT from extended criteria donors vs. liver resection | Single center; Norway | OS at 10 years | None listed |
| COLT | Nonrandomized, prospective | LT vs. triplet chemotherapy + anti-EGFR | Multicenter; Italy | OS at 5 years | PFS, complications |
| MELODIC | Nonrandomized, prospective | LT vs. chemotherapy | Multicenter; Italy | OS at 3 years, OS at 5 years | PFS, dropouts, complications |
| LIVERMORE | Single group, open label | LDLT vs. historical cohort of potentially transplantable patients who received chemotherapy only | Single center; Italy | OS at 5 years, DFS at 5 years | Graft survival, donor QOL |
| LITORALE2020 | Single group, open label | LT | Single center; Italy | OS at 5 years | DFS |
| NCT04874259 | Single group, open label | LDLT | Single center; Korea | OS at 1 year | DFS, OS at 3 years, recurrence |
| LIVERT(W)OHEAL | Single group, open label | LDLT with two-staged hepatectomy vs. historic cohort of patients who received gold-standard chemotherapy | Two centers; Germany | OS at 3 years | DFS, morbidity of recipient, morbidity of donor |

**Table 2.** *Cont.*

| Trial | Type | Design | Location | Primary Outcomes | Secondary Outcomes |
|-------|------|--------|----------|------------------|--------------------|
| RAPID-Padova | Single group, open label | LT with staged hepatectomy | Single center; Italy | Percent of patients receiving hepatectomy within 4 weeks of transplant | OS, PFS, dropouts, mortality, complications |
| TRASMETIR | Cohort, prospective | LT | Multicenter; Spain | OS at 5 years | DFS, QOL |
| NCT02864485 | Single group, open label | LDLT vs. patients who drop out prior to transplantation | Single center; Canada | OS at 5 years, DFS at 5 years | Recurrence, types of cancer recurrence treatments, dropouts, QOL, OS/DFS at 1 and 3 years |
| METLIVER | Single group, open label | LT | Multicenter; Spain | OS at 5 years | OS at 1 and 3 years, RFS, dropouts, recurrence, QOL |
| NCT06069960 | Single group, open label | Hemihepatectomy with concurrent left lateral lobe LT followed by delayed residual liver resection | Single center; China | OS at 3 years post second liver resection | DFS |
| RAPID 2014 | Single group, open label | Liver segmentectomy with concurrent left lateral lobe LT followed by delayed residual liver resection | Single center; Norway | Percent receiving second-stage hepatectomy within 4 weeks | OS |

Ongoing trials evaluating outcomes of liver transplantation for unresectable colorectal cancer metastases. From clinicaltrials.gov. LT—liver transplant; OS—overall survival; DFS—disease-free survival; PFS—progression-free survival; QOL—quality of life; TACE—transarterial chemoembolization; SIRT—selective internal radiation therapy; QALY—quality-adjusted life years; HAI/FUDR—hepatic arterial infusion with floxuridine.

At present, for many potential recipients of LT for unresectable CRLM, LDLT is the only option for transplant. LDLT is a known method to increase the organ pool and provide access to LT for patients who otherwise would not be able to attract a quality organ offer from the deceased donor list, but for many patients, this is not an option due to not having a suitable donor or not living near a center that offers this service. The problem with the use of marginal organs (a necessity at some centers with low-MELD recipients) is that it increases risk to an already high-risk operation. In addition, certain patients who are medically comorbid or who have complex arterial issues (the HAI pump adds complexity to the arterial inflow which may preclude LDLT) may not be able to tolerate a complicated postoperative course that comes with using a marginal allograft. These issues will need to be addressed to ensure equitable access to liver allografts for recipients who are likely to derive survival benefit from LT.

As discussed above, there are two phenotypes in patient populations who undergo LT for CRLM. Those who have unresectable disease and are highly selected for chronic disease tumor biology that has not spread outside the liver and have stable or treatment responsive disease and those whose liver is "burned out" from chemotherapy and locoregional therapies. These two patient populations should be studied independently and vigorously in order to determine who best will benefit from LT in the setting of CRLM.

A national and international registry needs to be created and made available to study. Societies such as the IHPBA and the International Liver Transplantation Society (ILTS) could

spearhead these efforts. This will allow the healthy and robust analysis of outcomes, which not only generates topics of further study but also allows for appraisal of centers, protocols, and techniques. Many questions remain regarding optimal preoperative imaging staging modalities, pre-LT chemotherapy regimens, and locoregional therapy options including use of the HAI pump, immunotherapy indications, and postoperative immunosuppression protocols. A unified database will make these issues easier to study and will allow for multicenter and multisociety collaboration.

## 8. Conclusions

In the past, LT for unresectable CRLM did not yield appropriate survival to justify the use of liver allografts (a severely limited resource) for this purpose. In the last two decades, however, several studies have defined selection criteria, which has led to favorable OS and DFS comparable to survival for other LT indications. While further study is needed, LT in this setting shows promise as a treatment modality for this disease.

**Author Contributions:** Conceptualization, T.J.R., K.C. and A.T.; literature review, T.J.R. and K.C.; tables and figures, T.J.R. and K.C.; writing—original draft preparation, T.J.R., K.C. and A.T.; writing—review and editing, T.J.R., K.C. and A.T. All authors have read and agreed to the published version of the manuscript.

**Funding:** This research received no external funding.

**Data Availability Statement:** No new data were created or analyzed in this study. Data sharing is not applicable to this article.

**Conflicts of Interest:** The authors declare no conflict of interest.

## Abbreviations

| | |
|---|---|
| CEA | carcinoembryonic antigen |
| CRC | colorectal carcinoma |
| CRLM | colorectal liver metastasis |
| DFS | Disease-free survival |
| ECOG | Eastern Cooperative Oncology Group |
| IHPBA | International Hepato-Pancreato-Biliary Association |
| ILTS | International Liver Transplantation Society |
| LT | liver transplant |
| OS | overall survival |
| RFS | recurrence-free survival |

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
