# Peer review of "Liver Transplantation for Unresectable Colorectal Liver Metastasis: Perspective and Review of Current Literature"

_curroncol, doi:10.3390/curroncol31020080_

Round 1

Reviewer 1 Report

Comments and Suggestions for Authors

Liver transplantation for colorectal liver metastases is undoubtely an emerging trend in transplant oncology.  As such, the number of reviews written during the last decade is vast, almost approaching the number of patients transplanted.  On this background, this particular review paper does not add anything new. It is very well written and the presentation of the l literature is succinct.

Some issues should be corrected/added/considered:

To be on par with the literature of LT for CRLM, the latest long-term report from the Oslo group should be considered: Dueland S, Smedman TM, Syversveen T, Grut H, Hagness M, Line P-D. Long-Term Survival, Prognostic Factors, and Selection of PatientsWith Colorectal Cancer for Liver Transplant. A Nonrandomized Controlled Trial. Jama Surg. 2023; doi: 10.1001/jamasurg.2023.2932.

Errors in reporting historical facts are important to correct. Otherwise, they tend to proliferate into other papers.  The authors claim that the first transplant for CRLM was performed by Aune et al (reference no. 14). This is not the case, and the authors must make sure that they report historical data correctly.  In fact, the idea of liver transplant as the ultimate liver resection for unresectable liver tumors is as old as the discipline of liver transplantation itself.  Starzl has given an overview of the first liver transplants in 1963 in the following paper; 

Starzl TE. The saga of liver replacement, with particular reference to the reciprocal influence of liver and kidney transplantation (1955-1967). Journal of the American College of Surgeons. 2002;195:587–610.

With regards to future implications:  Liver resection has been standard of care for CRLM.  Since the only alternative has been palliative oncological therapy, technical resectability has become the marker of "potential curative treatment".  With improved selection criteria in liver transplantation for CRLM, it is less obvious that liver resection is always the best treatment in patients with high tumor load, given that inferior survival and high recurrence rates have been consistently reported in CRLM with high tumor load (TBS ≥9 or more than 8-10 lesions) (Sasaki et al 2017, Allard et al 2017) and that a in TBS>9 there is no survival difference between R0 and R1 resections.

Thus,  the significance of the technical resectability of CRLM may in the future be a less obvious exclusion criterion for liver transplant than has been assumed so far.

Reference nr 22 is incomplete

Author Response

Liver transplantation for colorectal liver metastases is undoubtely an emerging trend in transplant oncology.  As such, the number of reviews written during the last decade is vast, almost approaching the number of patients transplanted.  On this background, this particular review paper does not add anything new. It is very well written and the presentation of the l literature is succinct.

Some issues should be corrected/added/considered:

To be on par with the literature of LT for CRLM, the latest long-term report from the Oslo group should be considered: Dueland S, Smedman TM, Syversveen T, Grut H, Hagness M, Line P-D. Long-Term Survival, Prognostic Factors, and Selection of PatientsWith Colorectal Cancer for Liver Transplant. A Nonrandomized Controlled Trial. Jama Surg. 2023; doi: 10.1001/jamasurg.2023.2932.

Thank you for this addition. Please see the paragraph starting at line 184 (highlighted).

Errors in reporting historical facts are important to correct. Otherwise, they tend to proliferate into other papers.  The authors claim that the first transplant for CRLM was performed by Aune et al (reference no. 14). This is not the case, and the authors must make sure that they report historical data correctly.  In fact, the idea of liver transplant as the ultimate liver resection for unresectable liver tumors is as old as the discipline of liver transplantation itself.  Starzl has given an overview of the first liver transplants in 1963 in the following paper; 

Starzl TE. The saga of liver replacement, with particular reference to the reciprocal influence of liver and kidney transplantation (1955-1967). Journal of the American College of Surgeons. 2002;195:587–610.

Thank you for this comment and correction. We have added this reference and made this change in the manuscript (see line 63 highlighted).

With regards to future implications:  Liver resection has been standard of care for CRLM.  Since the only alternative has been palliative oncological therapy, technical resectability has become the marker of "potential curative treatment".  With improved selection criteria in liver transplantation for CRLM, it is less obvious that liver resection is always the best treatment in patients with high tumor load, given that inferior survival and high recurrence rates have been consistently reported in CRLM with high tumor load (TBS ≥9 or more than 8-10 lesions) (Sasaki et al 2017, Allard et al 2017) and that a in TBS>9 there is no survival difference between R0 and R1 resections.

Thus, the significance of the technical resectability of CRLM may in the future be a less obvious exclusion criterion for liver transplant than has been assumed so far.

Thank you for this comment. The assessment of resectability is certainly always a moving target, especially with the evolution of surgical techniques, novel systemic therapy, as well as locoregional therapy. The article from the Oslo group we included (above) based on your suggestion identified TBS >9 and number of lesions >9 as being associated with inferior survival after LT. It is unclear at this point which therapies these patients with more aggressive tumor biology will benefit from.  

Reference nr 22 is incomplete

This has been corrected.

Reviewer 2 Report

Comments and Suggestions for Authors

The authors present a thoughtful update on LT for CRLM and present similar results as presented by Lee et al. 2022 in Cancer, Puia-Negulescu et al. 2021 in Int. J Mol. Science, Maspero 2023 in Cancers, Ros et al. 2023 in BJC, Lanari et al. 2020 in Current Transplantation reports. I do not think there is a lot of new information in this review, however I like the structure and the presentation. Due to the pressing nature of LT for CRLM I nevertheless think, this a valuable addition for the special issue transplant oncology of Current Oncology.

Points I would like to see adressed:

I would like to see table 2 be updated with more ongoing trials. Clinicaltrials.gov lists 17 active trials for LT for CRLM. You have listed 4.

Figure 1 could use some color to help the reader identify the flow chart. The orginal IHPBA figure is more concise.

Please discuss who you think should lead the effort in establishing "a national and international registry"

Minor details:

It should probably read Allan Tsung, MD not Allan Tsung and MD

Please make sure the above mentioned references are mentioned in you manuscript.

Author Response

The authors present a thoughtful update on LT for CRLM and present similar results as presented by Lee et al. 2022 in Cancer, Puia-Negulescu et al. 2021 in Int. J Mol. Science, Maspero 2023 in Cancers, Ros et al. 2023 in BJC, Lanari et al. 2020 in Current Transplantation reports. I do not think there is a lot of new information in this review, however I like the structure and the presentation. Due to the pressing nature of LT for CRLM I nevertheless think, this a valuable addition for the special issue transplant oncology of Current Oncology.

Points I would like to see adressed:

I would like to see table 2 be updated with more ongoing trials. Clinicaltrials.gov lists 17 active trials for LT for CRLM. You have listed 4.

Thank you. Please see updated table.

Figure 1 could use some color to help the reader identify the flow chart. The orginal IHPBA figure is more concise.

Please see color figure from IHPBA paper, included here with permission.

Please discuss who you think should lead the effort in establishing "a national and international registry"

This has been included in the manuscript, line 367.

Minor details:

It should probably read Allan Tsung, MD not Allan Tsung and MD

This has been corrected in author line.

Please make sure the above mentioned references are mentioned in you manuscript.

These are now included on line 61 (citation 14-19).